# Al_0.25_CoCrFeNiV High Entropy Alloy Coating Deposited by Laser Cladding on Stainless Steel

**DOI:** 10.3390/ma15207058

**Published:** 2022-10-11

**Authors:** Olga Samoilova, Nataliya Shaburova, Kirill Pashkeev, Marina Samodurova, Evgeny Trofimov

**Affiliations:** 1Department of Materials Science, Physical and Chemical Properties of Materials, South Ural State University, 76 Lenin Av., 454080 Chelyabinsk, Russia; 2Resource Center for Special Metallurgy, South Ural State University, 76 Lenin Av, 454080 Chelyabinsk, Russia

**Keywords:** high entropy alloys, gradient coatings, laser cladding, microstructure, microhardness

## Abstract

This paper studies the microstructure, composition and properties of a Al_0.25_CoCrFeNiV high entropy alloy coating (HEAC) deposited by laser cladding on austenitic-grade stainless steel. Laser cladding was carried out in an argon atmosphere on a FL-Clad-R-4 laser metal deposition complex with the following parameters: the laser power was 1400 W, the spot diameter was 3 mm, the track displacement was 1.2 mm, and the scanning speed was set to 10 mm/s. A change in the microstructure of the coating after laser cladding was revealed in comparison with as-cast high entropy alloy (HEA) Al_0.25_CoCrFeNiV. A significant decrease was found in the size of vanadium precipitates, from 20–40 µm in the as-cast state to 1–3 µm after laser cladding. A change in microhardness over the thickness of the coating from 370 HV_0.3_ at the outer surface to 270 HV_0.3_ at the boundary with the substrate was established, which may be due to the diffusion of Fe from the stainless steel into the coating material during laser cladding. Despite these features, the resulting coating adheres tightly to the substrate, and has no cracks or other defects, which indicates the possibility of using laser cladding to create coatings from high entropy alloys.

## 1. Introduction

Alloys consisting of five or more components in an equimolar ratio to achieve maximum mixing entropy were first studied by Cantor and Yeh in 2004 [1,2] and were called “high entropy alloys” (HEAs). Such alloys based on transition metals of the Fe subgroup (in particular, of the Al_x_CoCrFeNi type) demonstrate a unique combination of mechanical characteristics [3,4]; further studies showed their excellent tribological properties [5,6] and high corrosion resistance [7,8].

Recently, given the high cost of obtaining HEAs, an increasing amount of research is directed not to the manufacture of bulk products, but to the development of high entropy alloy coatings (HEACs). To obtain coatings from HEAs of the Al_x_CoCrFeNi type, various techniques have been used, such as electrospark deposition [9], gas tungsten arc welding (GTAW) [10,11,12], and thermal spraying [13,14,15]. These techniques have a number of disadvantages, such as the possible appearance of a Widmanstätt microstructure, high porosity of the coating, and the formation of large oxide particles in the coating volume.

At present, however, the most relevant research and production development is in the deposition of HEACs using laser cladding [16,17,18,19,20,21,22,23]. Despite a significant number of studies, there is no consensus on the most effective parameters of the process. Laser cladding of the base AlCoCrFeNi HEA was carried out by Zhang et al. [21] at a laser power of 2000 W; Liu et al. [20] changed to a laser power of 2400 W when Ti was added to the composition of this HEAC. Qiu et al. [18] increased the laser power to 2500 W to apply a coating of Al_2_CrFeCoCuTiNi_x_, a more complex composition. Samoilova et al. [23] carried out the laser cladding of Al_0.25_CoCrFeNiCu and Al_0.45_CoCrFeNiSi_0.45_ at much lower laser powers, 1600 and 1400 W, respectively. Moreover, Qiu et al. [18] and Liu et al. [20] found that the use of a high-power laser in the preparation of HEACs negatively affects the substrate material, since extensive (comparable to the coating thickness) heat-affected zones are created. Zhang et al. [21], Zhang et al. [22] and Samoilova et al. [23] indicate the diffusion of Fe from the steel matrix into the deposited coating, which changes its composition and potentially contributes to obtaining gradient properties over the thickness. In using laser cladding to apply HEACs, there are still large blind zones, and the development of the parameters is theoretically and practically significant.

The aim of this work is to develop a technique for applying HEACs using laser cladding. Al_0.25_CoCrFeNiV was chosen as the coating material because the addition of V increases the hardness and wear resistance of the base HEA AlCoCrFeNi [24,25,26] and positively affects the corrosion resistance in a sulfuric acid solution [27]. Considering that vanadium is prone to segregation into a separate phase, from our point of view, it is the use of laser cladding that should contribute to obtaining a defect-free coating with good adhesion to the matrix material. The microstructure, composition, and properties of the resulting Al_0.25_CoCrFeNiV HEAC were studied. It should be noted that there is no information in the literature on the use of a similar composition HEA as a coating material, which justifies the scientific novelty of the study.

## 2. Materials and Methods

Using the data from our previous work [23] on the uneven distribution of Al during laser cladding from a mixture of initial metal powders, the coating in this study was applied using a pre-melted HEA. For this purpose, an HEA ingot with the composition Al_0.25_CoCrFeNiV was smelted in an induction furnace according to the procedure in Shaburova et al. [28]. Then the ingot was subjected to crushing to obtain a powder with a fraction of 500 μm for fusing onto the substrate. The substrate material was austenitic grade stainless steel with the composition (at.%): 0.18 C, 1.10 Si, 0.46 Ti, 18.35 Cr, 1.63 Mn, 66.84 Fe, 9.97 Ni, 1.47 Mo.

The laser cladding of Al_0.25_CoCrFeNiV HEA was carried out in an argon atmosphere on the FL-Clad-R-4 laser metal deposition complex, the description of which is given in detail in [29,30]. In order to avoid the occurrence of an extensive heat-affected zone of the substrate material, the technological parameters of applying the Al_0.25_CoCrFeNiV coating in this study were as follows: the laser power was 1400 W, the spot diameter was 3 mm, the track displacement was 1.2 mm, and the scanning speed was 10 mm/s.

The study of the microstructure was carried out using a JSM-7001F scanning electron microscope (SEM) (JEOL, Tokyo, Japan) equipped with an energy dispersive spectrometer (EDS, Oxford Instruments, Abingdon, UK) for qualitative and quantitative X-ray spectral microanalysis. To analyze the grain boundaries, the coating material was etched in the marble reagent (20 g of copper sulfate, 100 mL of hydrochloric acid, 100 mL of distilled water); the substrate material was etched by the electrolytic method in a 10% aqueous solution of oxalic acid at a voltage of 5.5 V. The microstructure of thin sections after etching was studied using an Axio Observer D1.m optical inverted metallographic microscope (Carl Zeiss, Oberkochen, Germany) equipped with Thixomet Pro image analysis software and hardware. X-ray diffraction pattern analysis (XRD) of the samples was carried out on an Ultima IV diffractometer (Rigaku, Tokyo, Japan), the radiation used was CuK_α_. The microhardness of the coating and substrate was measured using an FM-800 microhardness tester (Future-Tech Corp., Tokyo, Japan) at a load of 300 g. The load holding time was 10 s. When measuring the microhardness of the samples, the load was chosen in such a way that the imprint fell on all microstructural components. For as-cast metal, measurements were carried out on at least twenty different points, the data were then averaged. For coating, hardness measurements were carried out along three lines at least 10–15 points in each; the results were also averaged.

## 3. Results and Discussion

### 3.1. Microstructure

The microstructure of the as-cast Al_0.25_CoCrFeNiV sample according to the SEM is shown in Figure 1a,b; the results of the EDS analysis are given in Table 1. In the as-cast sample in a dendritic (D) matrix (light gray phases in Figure 1a,b), a V-rich phase is present in the form of coarse dendrites and precipitates along grain boundaries (dark gray phases in Figure 1a,b). This is consistent with the data of [25,27], which indicate that V is prone to segregation into a separate phase.

The microstructure of the as-cladded Al_0.25_CoCrFeNiV HEA according to the SEM is shown in Figure 1c,d; the results of the EDS analysis are given in Table 1. After laser cladding, the microstructure of the alloy changed, there was a noticeable decrease in the size of the interlayers consisting of the V-rich phase from 20–40 µm to 1–3 µm. The morphology of V precipitations also changed, they are no longer dendritic, but needle-like or in the form of stars. This may be due to the accelerated heating, recrystallization, and rapid cooling during laser cladding. A similar change in the microstructure after laser cladding was also observed in our previous work [23] for Al_0.25_CoCrFeNiCu HEA.

In Figure 1d, single black dots no larger than 1 μm in size are distinguishable, which are the centers of the crystallization of the V phase. SEM and EDS studies showed that these are inclusions of Al_2_O_3_. This indicates oxygen contamination of the argon, and a possible suction of atmospheric oxygen during laser cladding. However, a detailed study of the coating did not show the accumulation of corundum inclusions, they are scattered and should not adversely affect the properties of the resulting HEAC.

According to optical microscopy, the thickness of the Al_0.25_CoCrFeNiV HEAC is about 1400–1500 μm (Figure 2a). There are no defects, pores, or cracks around the fusion of the coating material with the substrate material (Figure 2b), the thickness of the transition zone is about 50 µm, and elongated grains no more than 15 µm in length and 5 µm in width are distinguishable in the transition zone.

The microstructure of the HEAC after etching (Figure 2a,b) does not show porosity or cracks. Weld pools with different directions of crystallization, visible in the coating, indicate the recrystallization of metal areas adjacent to the laser beam. This may be due to the thickness of the HEAC layer and uneven heat removal during laser exposure. This microstructure differs favorably from the microstructure of the GTAW coating obtained by Chen et al. [11], where the authors indicate the presence of cracks, pores, and Widmanstätt structures, which may adversely affect the characteristics of coating. The defect-free coating obtained by laser cladding surpasses the HEACs deposited by thermal spraying. Thus, Tian et al. [14] directly point out that the high porosity of the deposited AlCoCrFeNiTi high entropy alloy coating is associated precisely with the features of the atmospheric plasma spraying (APS) process. On the other hand, the phenomenon of the formation of a large number of oxides particles in the volume of HEAC, obtained by the high velocity oxygen fuel (HVOF) process, draws the attention of Löbel et al. [15].

The substrate layer adjacent to the coating (Figure 2c) shows the absence of a heat-affected zone, which is apparently due to the low laser power (1400 W) used in laser cladding. A comparative analysis of the grain sizes in the boundary layer (Figure 2c) and in the body of the stainless steel matrix (Figure 2d) shows no differences: in both areas, the grain size does not exceed 70 μm and the microstructure is almost identical: equiaxed grains austenite with interlayers of ferrite (5–7%).

### 3.2. Distribution of Elements According to EDS

Taking into account the possibility of the diffusion of elements into the deposited coating [21,22,23,30], graphs of the change in the concentration of elements, based on EDS data, were plotted along the coating thickness and deep into the matrix to determine the distribution of elements (Figure 3a). It is clear that diffusion of Fe from the stainless steel into the coating takes place; on average, the concentration of Fe in the coating increased by a factor of two compared to the initial as-cast Al_0.25_CoCrFeNiV HEA (see Table 1). However, the graph in Figure 3a, shows the concentration gradient of Fe, V, and Co over the coating thickness. From the surface to the depth of the coating, the Fe concentration varies from 30 at.% up to 55 at.%, the concentrations of V and Co decrease from 16 at.% up to 9 at.%. Schematically, the process of saturation of the deposited coating with Fe ions is shown in Figure 4. With increasing distance from the substrate, the concentration of Fe in HEAC will more closely match the as-cast HEA, since the laser power and exposure time are not sufficient to completely saturate the coating with Fe ions throughout the entire thickness.

A similar significant saturation of the FeCoCrAlCu coating with the Fe of the matrix material during laser cladding was also discovered by Zhang et al. [22], who explain this phenomenon by the Marangoni effect in the liquid metal zone during laser cladding. According to Zhang et al. [22], this can be avoided by melting a preform for HEA without Fe, and only when applying the coating by laser cladding, carrying out a controlled saturation of the coating material with Fe. A significant disadvantage of this approach is the difficulty of obtaining the required concentration of Fe in the coating material.

### 3.3. Microhardness

The microhardness of as-cast Al_0.25_CoCrFeNiV HEA was 393 ± 10 HV_0.3_. When measuring the microhardness (Figure 3b), a gradient of HV_0.3_ along the coating thickness was found. The higher the Fe concentration (and the lower V and Co concentrations) in the coating, the lower the microhardness index HV_0.3_ (see Figure 3). The microhardness of the coating at a depth of 0–500 µm was 317–370 HV_0.3_, then the value decreased to 300 HV_0.3_, and in the transition zone, to 270 HV_0.3_.

We compared the data obtained in this study on the microhardness of the deposited coating with the data from Joseph et al. [31], where for the Al_0.3_CoCrFeNi alloy, the microhardness index was 176 HV_0.3_., and also with the data from Du et al. [32], according to which the microhardness for the base Al_0.25_CoCrFeNi HEA in the as-cast state was 151 HV_0.3_, and after annealing and rolling it was about 260 HV_0.3_. Thus, the investigated coating, despite the observed Fe diffusion, has a high microhardness index due to the addition of V to the base composition.

### 3.4. XRD Analysis

Results of X-ray phase analysis are given in Figure 5.

The substrate is the fcc phase, which we define as austenite (see Figure 5a). Due to the small amount of the ferrite phase, it is not determined on the diffraction pattern. According to Figure 5b, the as-cast HEA is characterized by the presence of an fcc solid solution and bcc precipitates of a vanadium-enriched (V-rich) phase. After laser cladding, the peaks of the V-rich phase in the diffraction pattern become almost imperceptible, which coincides with the observed microstructure (see Figure 1). Thus, laser cladding promotes the transition of V into a solid solution and reduces the percentage of precipitation of dendrites in the bcc phase.

For high entropy alloys, it is customary to indicate the calculated values of the entropy of mixing. According to Boltzmann’s theory [33]
$$\Delta S_{\mathrm{mix}}=-R\sum_{i=1}^{n}X_{i}\ln X_{i}$$
where *R* is the universal gas constant (*R* = 8.314 J∙mol^–1^∙K^–1^) and *X* is the atomic fraction of the element in the HEA.

Using data from Table 1, it is possible to calculate the entropy of mixing for each of the phases in Figure 5b. The results of the calculation are given in Table 2.

Table 2 shows that the as-cast Al_0.25_CoCrFeNiV alloy can be described as a composite material in which a low-entropy strengthening phase is located in a high-entropy matrix. After laser cladding, due to the diffusion of Fe from the stainless steel into the coating material, the entropy of mixing for the fcc solid solution in the HEA decreased but remained at the values ≥ 1.5R for high-entropy phases. Of course, the calculation of the entropy of mixing for fcc solid solution in the deposited coating is an estimate and was calculated for the average composition from Table 1. The coating turned out to be gradient in Fe concentration (see Figure 3a), which means that each of these layers in the coating has its own calculated value of the entropy of mixing. Apparently, for a transition layer near the substrate, the entropy of mixing can be lower than necessary, and such a layer in the coating can be attributed to a medium (low) entropy alloy. On the other hand, laser cladding not only improved the composition and morphology of the enriched V phase, but also increased its value entropy of mixing.

## 4. Conclusions

The results show that it is possible to obtain gradient coatings from high entropy alloys using laser cladding. The use of a relatively low laser power (1400 W) in the deposition of the Al_0.25_CoCrFeNiV HEAC made it possible to avoid the occurrence of a heat-affected zone in the matrix material (stainless steel), but did not prevent the diffusion of Fe from the matrix into the coating. The presence of Fe diffusion is the main factor for obtaining a coating with a gradient in composition and microhardness, when sufficiently hard outer layers of the coating (370 HV_0.3_) pass into microhardness values 270 HV_0.3_ close to the substrate material. Accelerated heating, recrystallization, and rapid cooling under laser irradiation also improved the microstructure of Al_0.25_CoCrFeNiV HEA: a significant decrease in the size of vanadium precipitates from 20–40 µm for the as-cast state to 1–3 µm in the coating obtained by laser cladding was revealed.

## Figures and Tables

**Figure 1 materials-15-07058-f001:**
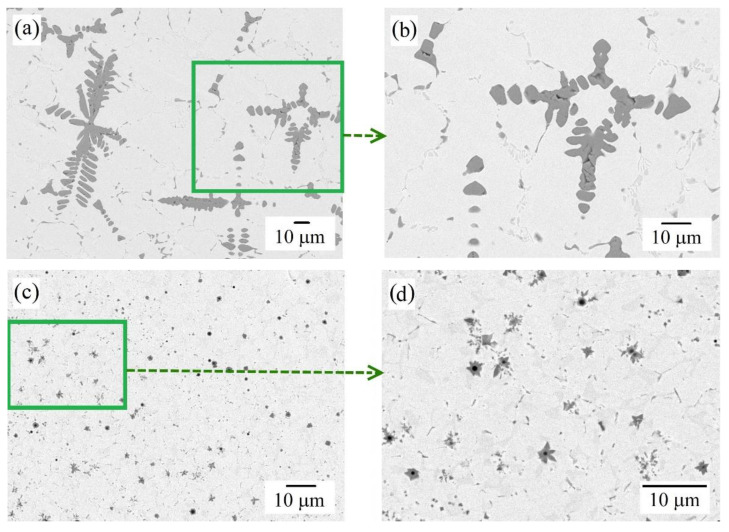
The Al_0.25_CoCrFeNiV HEA microstructure according to SEM: (**a**,**b**) in as-cast state; (**c**,**d**) in as-cladded state.

**Figure 2 materials-15-07058-f002:**
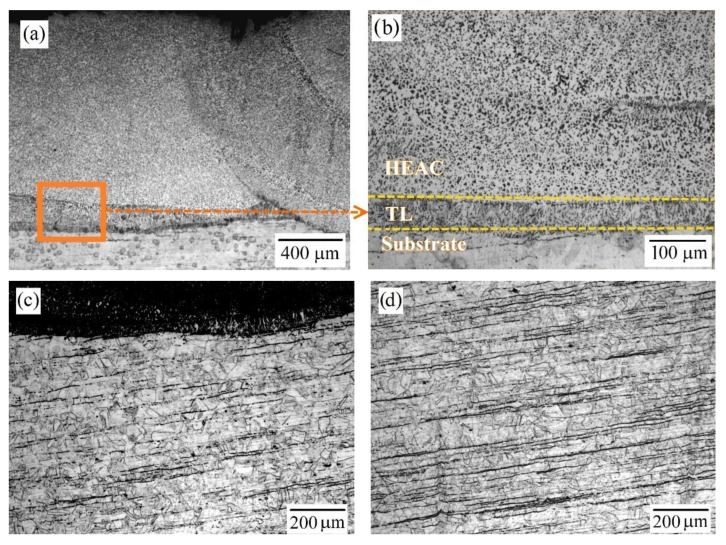
The microstructure after etching according to optical microscopy: (**a**) general view; (**b**) the fusion area of the coating material with the substrate material; (**c**) substrate layer bordering the coating; (**d**) microstructure in the substrate body. TL—transition layer.

**Figure 3 materials-15-07058-f003:**
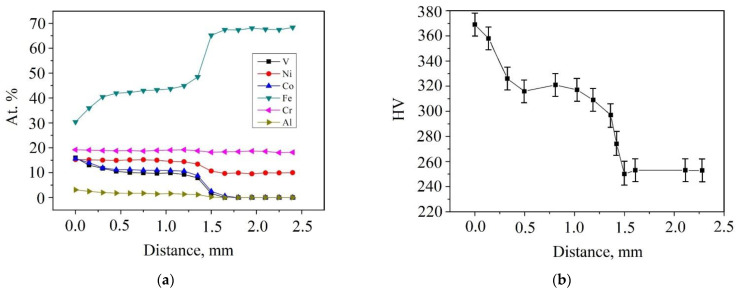
Distribution over the thickness of the coating and substrate: (**a**) concentrations of elements according to EDS; (**b**) microhardness.

**Figure 4 materials-15-07058-f004:**
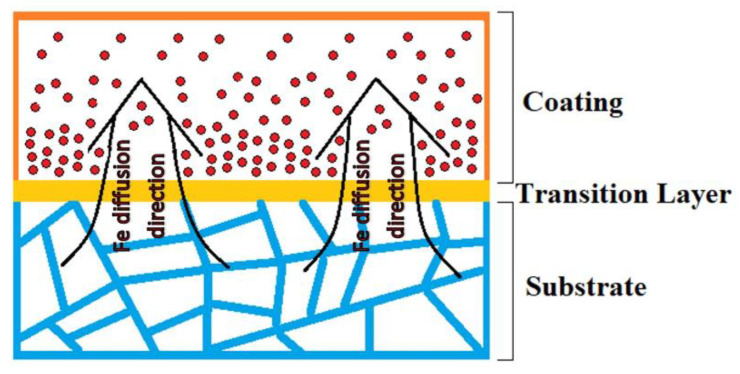
Scheme of coating saturation with Fe ions.

**Figure 5 materials-15-07058-f005:**
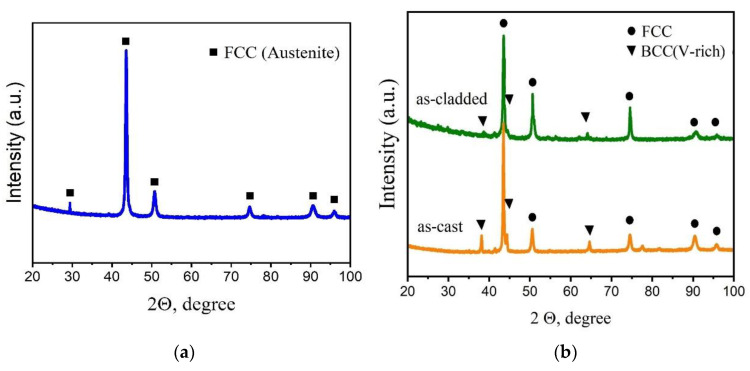
X-Ray diffraction patterns: (**a**) substrate; (**b**) Al_0.25_CoCrFeNiV HEA for as-cast and as-cladded states.

**Table 1 materials-15-07058-t001:** Chemical compositions according to EDS (at.%): Av—average composition, D—dendrite, V-rich—interdendritic enriched V region.

Alloy		Al	Cr	Fe	Co	Ni	V	Si	Mn	Mo	Ti
Al_0.25_CoCrFeNiV in as-cast state	Av	4.69	19.26	18.98	19.01	18.94	19.12	-	-	-	-
	D	4.34	18.96	21.15	21.35	21.52	12.68	-	-	–	-
	V-rich	0.87	8.78	1.45	1.49	1.78	85.63	-	-	-	-
Al_0.25_CoCrFeNiV in as-cladded state	Av	1.44	18.98	42.69	10.61	15.18	9.29	0.58 *	0.51 *	0.64	0.08 *
	D	1.29	18.31	45.06	10.20	15.53	7.63	0.56 *	0.81	0.61	–
	V-rich	1.10	15.41	21.10	8.22	8.56	43.23	-	-	-	2.38

*—The determination error for low concentrations of elements (≤0.5 at.%) is comparable with the determined values, therefore, these data should be considered not as quantitative, but as qualitative values that determine the main trend in the diffusion behavior of elements.

**Table 2 materials-15-07058-t002:** The entropy of mixing for each of the phases of Al_0.25_CoCrFeNiV HEA in as-cast and as-cladded states.

Phase	As-Cast State		As-Cladded State	
	D (FCC)	V-Rich (BCC_V-rich_)	D (FCC)	V-Rich (BCC_V-rich_)
$\Delta S_{\mathrm{mix}},J\cdot {\mathrm{mol}}^{-1}\cdot K^{-1}$	14.15	4.85	12.83	12.75

## Data Availability

The data presented in this study are available on request from the corresponding author.

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
