# Peer review of "Al0.25CoCrFeNiV High Entropy Alloy Coating Deposited by Laser Cladding on Stainless Steel"

_materials, 2022, doi:10.3390/ma15207058_

Round 1

Reviewer 1 Report

The authors studied the microstructure, composition and properties of HEAC after laser cladding on stainless steel. The article focuses more on testing and less on experimental variables. There are many laser variables, it is suggested to study the influence of different variables on the structure and properties of materials. The following details need to be revised.

1. Please add stainless steel substrate to the title.

2. What is the innovation of this article? If the laser cladding process is preferred, the design of process variables is very necessary. Why choose this kind of high entropy alloy? What are its main application areas? It is recommended that the material characteristics described in the last paragraph of Part 1 be moved to the first paragraph.

3. Is the percentage of stainless steel composition by mass or volume? What is the holding time of the load in microhardness testing?

4. Please indicate the position of each phase in Figure 1. Is Fig. 1(b) a partial enlargement of Fig. 1(a)? The corresponding area must be indicated. The same modification is required for Fig. 1(c) and Fig. 1(d).

5. Please provide a schematic diagram of the experimental process.

6. Please adjust the dimensions of each figure in Fig. 2 to be consistent. In addition, it is necessary to distinguish each area with a line and indicate whether it is substrate or transition layer or HEAC. The grain in Fig. 2 also need to be marked.

7. Please provide a schematic diagram of the elemental changes in Part 3.2.

8. In Fig. 3, where is the test location? For EDS testing, it is recommended to use line scanning to indicate the change trend of elements.

9. Have XRD tests at different thicknesses been considered? For as-cladded sample, what are the phases from 25 degree to 35 degree? Because the phases are disappeared at the same degree of as-cast sample.

Author Response

Please find the answer in the attached file.

Reviewer 2 Report

It is a very interesting contribution reporting on the use of laser cladding to produced a good-quality HEA coating for which the particular effect of V addition is investigated. There are some minor issues that must be addressed before publishing.

Comments

1. Include a very recent review on laser cladding of HEA together with Refs. [16-23]:

High Entropy Alloys Coatings Deposited by Laser Cladding: A Review of Grain Boundary Wetting Phenomen. Coatings  12(3) (2022) 343. DOI: 10.3390/coatings12030343

2. The accuracy of the EDS results can never be 0.01 %. Even with standard the best can be 0.5 %. Therefore these data must be corrected accordingly.

3. Which was the size of the indentation? The as-cast alloy is not homogeneous.

4. It is impossible to identify a phase from only two peaks in the XRD pattern. Please, correct it.

5. Since the Fe content in the coating is not homogeneous to talk about a mixing enthalpy for which this value is assumed constant is risky. In addition, this is used in the conclusions. Please, clarify.

Author Response

First of all, we want to thank the Reviewer for his constructive comments, which helped to significantly improve the text of the manuscript. We carefully read the comments and tried to fully respond to them.

All corrections in the text of the manuscript are highlighted with a green marker.

Comments

  1. Include a very recent review on laser cladding of HEA together with Refs. [16-23]:

High Entropy Alloys Coatings Deposited by Laser Cladding: A Review of Grain Boundary Wetting Phenomen. Coatings  12(3) (2022) 343. DOI: 10.3390/coatings12030343

Answer: Corresponding changes have been made to the text of the manuscript.

  1. The accuracy of the EDS results can never be 0.01 %. Even with standard the best can be 0.5 %. Therefore these data must be corrected accordingly.

Answer: Yes, according to the theoretical foundations, the sensitivity of the EDS is tenths of a percent. However, we use a modern detector with an area of 80 mm2 (INCA X-MAX), which allows us to hope for a higher sensitivity of the method. Moreover, we used the Aztec software, which allows us to calculate difficult-to-distinguish peaks. The numbers given in Table 1 are calculated by the program, the error is calculated for them too. For small concentrations of elements the error of determination is large, which makes it possible to use these data more as qualitative than as quantitative. That is why in Fig. 3 we did not give the distribution for elements with low concentration (Mn, Mo, Ti, etc.).

Corresponding corrections were made to the text of the manuscript.

  1. Which was the size of the indentation? The as-cast alloy is not homogeneous.

Answer: Corresponding explanations are included in the text of the manuscript.

“When measuring the microhardness of the samples, the load was chosen in such a way that the imprint fell on all microstructural components. For as-cast metal, measurements were carried out on at least twenty different points, the data were then averaged. For coating, hardness measurements were carried out along three lines at least 10–15 points in each; the results were also averaged.”

  1. It is impossible to identify a phase from only two peaks in the XRD pattern. Please, correct it.

Answer: Corresponding corrections were made to the text of the manuscript.

  1. Since the Fe content in the coating is not homogeneous to talk about a mixing enthalpy for which this value is assumed constant is risky. In addition, this is used in the conclusions. Please, clarify.

Answer: Corresponding explanations are included in the text of the manuscript.

“Of course, the calculation of the enthalpy of mixing for fcc solid solution in the deposited coating is an estimate and was calculated for the average composition from Table 1. The coating turned out to be gradient in Fe concentration (see Fig. 3a), which means that each of these layers in the coating has its own calculated value of the enthalpy of mixing. Apparently, for a transition layer near the substrate, the entropy of mixing can be lower than necessary, and such a layer in the coating can be attributed to a medium (low)-entropy alloy.”

Reviewer 3 Report

The following are my review comments;

Abstract

The abstract is well formulated, but a few punctuation errors must be eliminated for better readability.

In line 12, specify the type of stainless steel used as substrate.

In line 15, “A change in microstructure..”, try to split the sentence and specify the vanadium precipitates as a different sentence.

Introduction section

Throughout the manuscript, give citations at the end of the sentence, giving readers good continuity.

Briefly review the coatings techniques mentioned in lines 36 and 37.  Moreover, suggest why this study was carried out in laser cladding and its importance rather than mentioning “no consensus in the process parameter…” as per line 40.

Results and Discussion

From the materials and methods section, it is quoted that the etchant used is a Marble reagent.  Is the same etchant used for taking microstructure for the coated region, fusion region, substrate layer bordering the coating, and substrate body?  Give a detailed explanation regarding the metallographic procedure in the materials and methods section.

From lines 121 to 124, the authors have compared the microstructure of the present study with that of GTAW.  It is a well-known fact that during GTAW, cracks, pores and Widmanstatt structures within the permissible range are not considered a defect as per weld standards.  If so, why do the authors have to compare the present study with GTAW, and why not compare it with other techniques also, as mentioned in line 36 (i.e. electro spark deposition and thermal spaying)

 In line 132, the authors have mentioned the grain size of less than 70µm.  Justify by plotting grain size histogram graph.

From line 162, compare data obtained from the present work with the other data from Joseph et al. and Du et al.  Give justification for your findings also.

Provide the XRD graph of the substrate also.  This gives a better understanding of the type of stainless steel used.

 The manuscript may be accepted based on incorporating the above revisions.

Author Response

First of all, we want to thank the Reviewer for his constructive comments, which helped to significantly improve the text of the manuscript. We carefully read the comments and tried to fully respond to them.

All corrections in the text of the manuscript are highlighted with a blue marker.

1) Abstract

The abstract is well formulated, but a few punctuation errors must be eliminated for better readability.

Answer: We rechecked the text of manuscript again.

2) In line 12, specify the type of stainless steel used as substrate.

Answer: Austenitic grade stainless steel was used as the substrate.

Corresponding corrections were made to the text of the manuscript.

3) In line 15, “A change in microstructure..”, try to split the sentence and specify the vanadium precipitates as a different sentence.

Answer: Corresponding corrections were made to the text of the manuscript.

Introduction section

4) Throughout the manuscript, give citations at the end of the sentence, giving readers good continuity.

Answer: The manuscript was prepared in accordance with the instructions for MDPI authors.

5) Briefly review the coatings techniques mentioned in lines 36 and 37.  Moreover, suggest why this study was carried out in laser cladding and its importance rather than mentioning “no consensus in the process parameter…” as per line 40.

Answer: Corresponding corrections were made to the text of the manuscript.

“These techniques have a number of disadvantages, such as the possible appearance of a Widmanstätt microstructure, high porosity of the coating, and the formation of large oxides particles in the coating volume.”

“Considering that vanadium is prone to segregation into a separate phase, from our point of view, it is the use of laser cladding that should contribute to obtaining a defect-free coating with good adhesion to the matrix material.”

Results and Discussion

6) From the materials and methods section, it is quoted that the etchant used is a Marble reagent.  Is the same etchant used for taking microstructure for the coated region, fusion region, substrate layer bordering the coating, and substrate body?  Give a detailed explanation regarding the metallographic procedure in the materials and methods section.

Answer: Corresponding corrections were made to the text of the manuscript.

“To analyze the grain boundaries, the coating material was etched in the Marble reagent (20 g of copper sulfate, 100 ml of hydrochloric acid, 100 ml of distilled water); the substrate material was etched by the electrolytic method in a 10% aqueous solution of oxalic acid at a voltage of 5.5 V.”

7) From lines 121 to 124, the authors have compared the microstructure of the present study with that of GTAW.  It is a well-known fact that during GTAW, cracks, pores and Widmanstatt structures within the permissible range are not considered a defect as per weld standards.  If so, why do the authors have to compare the present study with GTAW, and why not compare it with other techniques also, as mentioned in line 36 (i.e. electro spark deposition and thermal spaying).

Answer: Corresponding corrections were made to the text of the manuscript.

“The defect-free coating obtained by laser cladding surpasses the HEACs deposited by thermal spraying. Thus, Tian et al. [14] directly point out that the high porosity of the deposited AlCoCrFeNiTi high entropy alloy coating is associated precisely with the features of the atmospheric plasma spraying (APS) process. On the other hand, the phenomenon of the formation of a large number of oxide particles in the volume of HEAC, obtained by the high velocity oxygen fuel (HVOF) process, draws the attention of Löbel et al [15].”

8)  In line 132, the authors have mentioned the grain size of less than 70µm.  Justify by plotting grain size histogram graph.

Answer: The grain size was determined using the ThixometPro software-analytical complex for image analysis by comparison with standard scales. This method allows to set the austenite grain score and its corresponding maximum grain size in µm. Due to the fact that there is no gradient in the size of austenite grains in the substrate after laser cladding of coating, we considered that the grain size distribution histogram does not carry any semantic load, does not affect the results obtained, and will only overload the manuscript.

9) From line 162, compare data obtained from the present work with the other data from Joseph et al. and Du et al.  Give justification for your findings also.

Answer: Corresponding corrections were made to the text of the manuscript.

“We compared the data obtained in this study on the microhardness of the deposited coating with the data from Joseph et al. [31], where for the Al0.3CoCrFeNi alloy, the microhardness index was 176 HV0.3. And with the data from Du et al. [32], according to which the microhardness for the base Al0.25CoCrFeNi HEA in the as-cast state was 151 HV0.3, and after annealing and rolling it was about 260 HV0.3. Thus, the investigated coating, despite the observed Fe diffusion, has a high microhardness index due to the addition of V to the base composition.”

10) Provide the XRD graph of the substrate also.  This gives a better understanding of the type of stainless steel used.

Answer: Corresponding corrections and additions were made to the text of the manuscript (figure added).

“The substrate is the fcc phase, which we define as austenite (see Fig. 5a). Due to the small amount of the ferrite phase, it is not determined on the diffraction pattern.”

Round 2

Reviewer 1 Report

It is recommended to accept and publish this article.

Reviewer 3 Report

Accept in the present format